# Flexibility-tuning of dual-display DNA-encoded chemical libraries facilitates cyclic peptide ligand discovery

Dimitar Petrov [1], Louise Plais [1], Kristina Schira [1], Junyu Cai [1], Michelle Keller[1], Alice Lessing[1], Gabriele Bassi[2], Samuele Cazzamalli [2], Dario Neri[2], Andreas Gloger [1] & Jörg Scheuermann [1] ✉

Cyclic peptides constitute an important drug modality since they offer significant advantages over small molecules and macromolecules. However, access to diverse chemical sets of cyclic peptides is difficult on a large library scale. DNA-encoded Chemical Libraries (DELs) provide a suitable tool to obtain large chemical diversity, but cyclic DELs made by standard DEL implementation cannot efficiently explore their conformational diversity. On the other hand, dual-display Encoded Self-Assembling Chemical (ESAC) Libraries can be used for modulating macrocycle flexibility since the two displayed peptides can be connected in an incremental fashion. In this work, we construct a 56 million dual-display ESAC library using a two-step cyclization strategy. We show that varying the level of conformational restraint is essential for the discovery of specific ligands for the three protein targets thrombin, human alkaline phosphatase and streptavidin.

The discovery and development of peptide-based drugs, especially constrained macrocyclic peptides, has recently gained momentum[1–6]. These molecules of intermediate size promise combining the advantages of both the high specificity of antibodies, and the better tissue and cell permeability of small molecules[7–9].

Currently, biological display libraries of peptides, such as phage- and mRNA-display have proven highly efficient at isolating ligands for a wide range of targets[10–14]. These libraries can be very large-sized and typically yield high-affinity ligands if the peptides are appropriately rigidified in a macrocyclic structure. However, these display technologies rely on ribosomal translation-compatible amino acid building blocks and typically afford hits of larger ring sizes, which might require substantial optimization for drug development, due to often poor cell-permeability and in vivo stability[2,14,15].

Recent approaches relying on solid phase peptide synthesis (SPPS) succeeded in expanding the repertoire of available chemistries and building blocks to yield more drug-like macrocycles. Such approaches include the picomole scale macrocycle platform developed by C. Heinis and coworkers where small drug-like macrocycles

are synthesized and screened in an arrayed high-throughput setting[4,16,17]. While such a platform produces much more druglike hits, it is naturally limited in reaching the vast numbers of encoded display technologies. Furthermore, the suprabody technology developed by N. Winnsinger and coworkers has elegantly enabled the generation of large libraries of encoded PNA-peptide conjugates which can combinatorially assemble into complexes on-DNA[18]. While not forming fully cyclic products the conformation of these assembled structures could be constrained through hybridization of flanking nucleic acid tags.

On the other hand, DNA-encoded chemical libraries (DEL) leverage the vast options offered by synthetic combinatorial chemistry for the creation of large collections of small molecules, where each member is coupled to a unique DNA sequence[19–30]. DEL technology allows for the incorporation of a large array of building blocks through diverse chemical reactions, rendering it a cornerstone in small molecule drug discovery[31–41]. Cyclic peptide DELs have been constructed which benefit from the vast diversity of unnatural amino acids that can be incorporated[42–52]. However, currently practiced DEL technology

[1]Institute of Pharmaceutical Sciences, Department of Chemistry and Applied Biosciences, ETH Zurich, Vladimir-Prelog-Weg 3, 8093 Zurich, Switzerland. [2]Philochem AG, Libernstrasse 3, 8112 Otelfingen, Switzerland. ✉e-mail: joerg.scheuermann@pharma.ethz.ch

faces several challenges with respect to synthesizing larger peptides: synthetic limitations restrict cyclic peptide DELs to a low number of diversity element positions (typically less than four) compared with their natural display counterparts[53–56], and typically produce much smaller (cyclic) peptide structures, with the resulting peptides being either conformationally restrained if cyclized or conformationally flexible if linear, i.e., without an option for fine-tuning. As minor adjustments in cycle flexibility can have major effects on the binding affinity, especially for smaller macrocycles, a proper balancing between flexibility and structural rigidity is required[57–59]. While the flexibility of cyclic peptides might, to some extent, be controlled by the use of diverse sets of building blocks, which might additionally influence flexibility through intramolecular interactions in an unpredictable way, a higher degree of flexibility control implemented directly into the library design may allow for the optimal fine-tuning of cyclic structures to best fit into a target protein's binding pocket.

In this work, we adapted dual-display Encoded Self-Assembling Chemical (ESAC) library technology which had formerly been developed for the discovery of simultaneously binding small fragments[60–68] to the synthesis of flexibility-tunable macrocycles. We leveraged our recently developed Large Encoding Design (LED) to an enhanced ESAC encoding system, since it permits to simultaneously encode two or more building blocks of the sub-libraries on one DNA strand which can eventually be amplified after affinity-based selection and sequenced[69,70]. This allowed us to combinatorially assemble two peptidic sub-libraries. The setup divided the synthetic work into constructing the two sub-libraries, each made-up by two synthetic cycles, thereby rendering the final ESAC-DEL much purer compared with performing four consecutive synthesis cycles on one DNA strand. Importantly, the ESAC-DEL could contain three levels of rigidification described as open, semi-closed and closed arrangements (see Fig. 1), which permitted the isolation of ligands against various target proteins. By performing affinity-based selections, we identified specific preferences for the various degrees of flexibility: we isolated a 314 nM thrombin inhibitor from the conformationally restrained macrocyclic subset of the ESAC library, while streptavidin preferred a more flexible semi-closed conformation. In contrast, human alkaline phosphatase (PLAP) ligands were discovered only from the open, conventional dual-display setup. Our findings demonstrate that, by incorporating three levels of peptide rigidity within one ESAC library, dual-display technology facilitates the de novo discovery of peptide ligands through flexibility-tuning.

## Results

### Library construction

Encoded Self-Assembling Chemical (ESAC) libraries feature the display of two pharmacophores on the 5'- and the 3'-ends of two complementary DNA strands, respectively. Sub-libraries of the two individual strands can be produced separately and later hybridize to form a combinatorial DEL of stable DNA-heterodimers[60,61]. The Y-shaped Large Encoding Design (LED) methodology enabled the proper encoding by eventually obtaining a unique DNA strand that encodes for all the building blocks composing the heterodimer as well as the different versions of the DP-DEL (open, semi-closed, closed)[61,69]. The sub-libraries both contained a cysteine scaffold, onto which two sequential couplings of amino acid diversity elements took place, as well as a conjugation of an azide and an alkyne functionality, respectively[69,71,72]. After pairing of the two sub-libraries the two peptides could subsequently be linked and conformationally restricted by joining them at their N- and C-termini (Fig. 1). First, a copper-catalyzed click reaction was used for the first step of rigidification. The two peptides were covalently connected at their N-terminus through a triazole ring, while also being restrained via the hybridization of the two DNA strands. The C-termini of the displayed peptides could subsequently also be linked in a second rigidification step to form a complete macrocycle (Fig. 1). While the rigidification-tuning via bis-electrophile ring-closure was intrinsically implemented in the design of the closed version of the DP-DEL, for both the open and semi-closed versions of the DP-DEL a fine-tuning of rigidity was accomplished post-selection during hit validation using linkers of varying lengths and flexibility.

The ESAC design also enabled us to improve upon library purity. After the first step of the split-and-pool synthesis, the sub-library members could still be individually purified and characterized, such that only the second reaction steps were performed in mixtures. Therefore, the overall synthesis amounts to the construction of two quite pure DEL sub-libraries with two diversity elements each, in contrast to a macrocyclic DEL of four diversity elements prepared by conventional DEL technology, which is highly prone to variable yields[56]. The incorporation of pre-synthesized dipeptide building blocks allowed us to also explore larger ring-sizes while retaining the same level of synthetic purity. Moreover, the N-N and C-C cyclization might be facilitated by the high effective molarity due to DNA templating[73].

The synthesis for each sub-library began with a single-stranded amino-modified oligonucleotide (**HP5** for the 5' strand and **HP3** for the respective 3' strand (Fig. 1A)). Both oligonucleotides were first coupled to Fmoc-L-Cys(oNv)-OH scaffold, whose photo-deprotection was specifically optimized[74](Supplementary Information, section 3.2 Fig. 1, step I). After Fmoc group removal, the Cys-modified sub-libraries were split and reacted with a number of amino acid building blocks (175 BBs for the **HP5** and 176 BBs for the **HP3** sub-library, depicted as BB1 and BB3 in Fig. 1), selected from a panel of natural and unnatural amino acids, to maximize side chain diversity. In addition, 78 dipeptides of natural and unnatural origin were individually synthesized and included for enlarging the final ring size and flexibility. Subsequently, all conjugates were Fmoc-deprotected, HPLC purified, and reactions were encoded via adaptor-mediated ligation (Fig. 1, step II). The individual conjugates were then combined into Pool 1 for **HP5** and **HP3** sub-libraries, respectively.

Library synthesis proceeded for **HP5** sub-library with the installation and subsequent encoding of 152 amino acid BB2 by amidation (Fig. 1, step III). The reactions were then Fmoc-deprotected and subjected to diazotransfer. For the **HP3** sub-library, installation of BB4 involved splitting the **HP3** Pool 1 into two portions, followed by encoding via adaptor-mediated ligation and amidation using two different carboxylic acid alkynes (Fig. 1, step IV) and a final piperidine treatment.

The partially complementary **HP5** and **HP3** sub-libraries were then mixed in equimolar ratios and allowed to hybridize, resulting in the open **version 1** of DP-DEL with a total of 9,363,200 members (Fig. 1). This configuration resembles the traditional implementation of dual-display ESAC, where displayed library members are unlinked. As the N-terminus of each peptide sub-library carried an alkyne and an azide group, respectively, while the C-terminus carried two protected cysteines, the two sub-libraries could further be covalently linked and assembled into a macrocycle in a step-wise manner: the first conformational restriction was introduced by reacting the azide and alkyne of the **HP5** and **HP3** sub-libraries by copper-catalyzed Huisgen cycloaddition (CuAAC). This resulted in the semi-closed **version 2** of DP-DEL, where the two peptides are covalently linked at their N-terminus, while also being held in proximity at the C-termini via the hybridization of the two DNA strands. Final derivatization of the DP-DEL was obtained by deprotection of the Cys-oNv C-termini, allowing for the formation of a disulfide between the cysteines, thus yielding complete macrocyclic structures. The cysteines could additionally also be reduced to react with three bifunctional electrophiles, offering additional degrees of flexibility in cycle formation, yielding closed DP-DEL **version 3** (Fig. 1). Our DEL synthesis hence afforded three versions of ESAC libraries: unrestricted peptides (**version 1**), peptides reacted at their N- termini (**version 2**), as well as fully closed macrocycles,

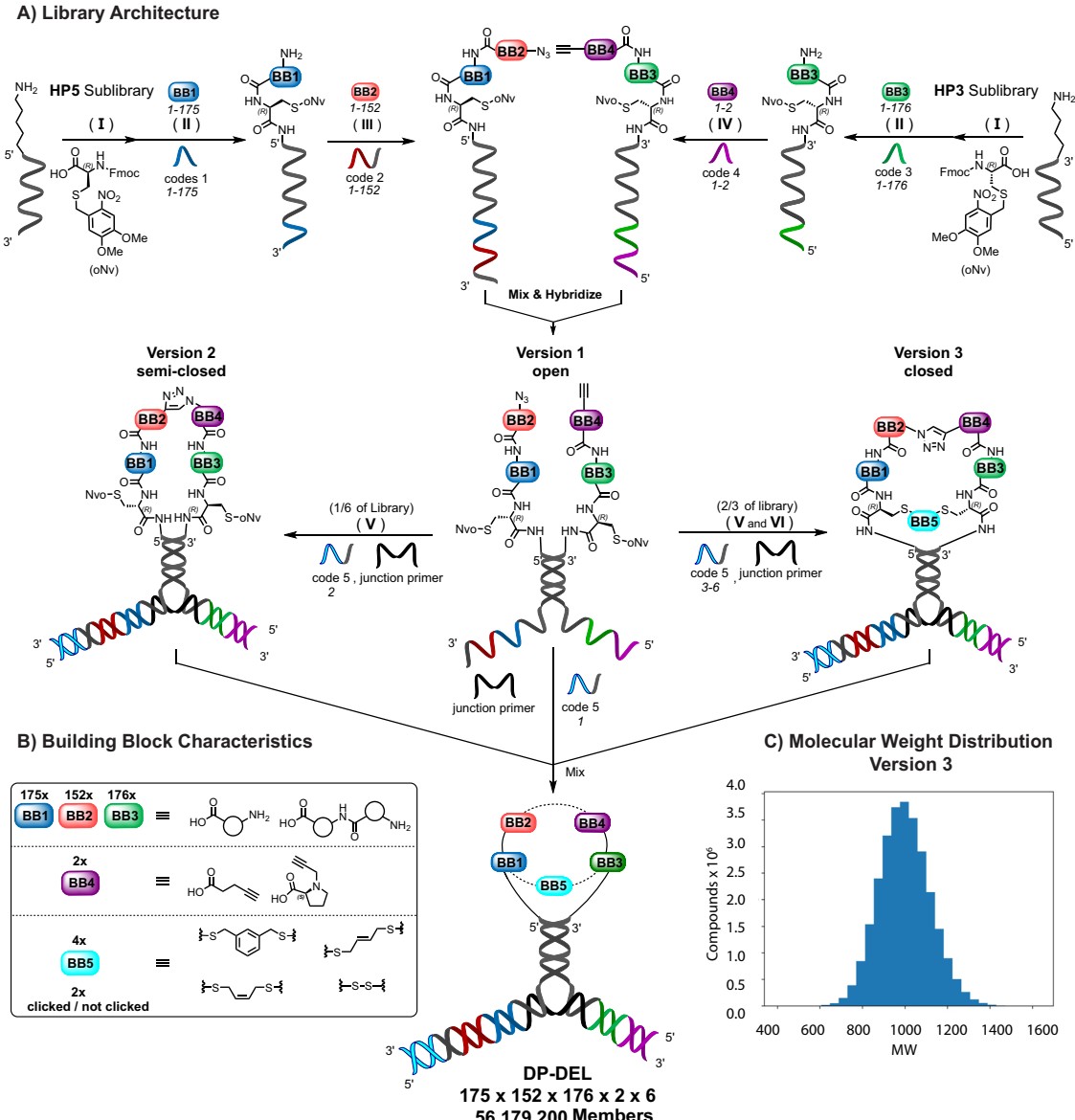

**Fig. 1 | DP-DEL library architecture and synthesis. A** Schematic representation of DP-DEL synthesis. The building blocks (BB1-5) and their corresponding DNA codes are color-coded. The library is formed by the hybridization of the synthesized sub-libraries **HP5** and **HP3** and optional crosslinking. Fmoc-L-Cys(oNv)-OH was conjugated as a scaffold to both partially complementary 5′ NH$_2$-modified 38-mer HP5 and 3′ NH$_2$-modified 38-mer HP3, followed by Fmoc group removal (**I**). BB1 and BB3 were conjugated via amide bond formation onto the scaffolds of HP5 and HP3 strands, respectively, followed by Fmoc removal and HPLC purification of each conjugate. Enzymatic adaptor-mediated ligation was performed for the encoding of each building block, before pooling **HP5** and **HP3** conjugates, respectively (**II**). The HP5 strand was sequentially reacted with BB2 which were attached using amide bond formation, followed by enzymatic adaptor-mediated ligation, Fmoc deprotection and diazo-transfer onto the free amino function (**III**). The HP3 strand was pre-encoded with BB4 codes, followed by amide bond formation of BB4 and piperidine treatment to remove DMT-MM adducts (**IV**). **HP5**- and **HP3**-conjugate pools were HPLC purified, then the two sub-libraries were mixed stoichiometrically, allowing the **HP5** and **HP3** strands to hybridize, resulting in **version 1** of DP-DEL. The major part of the hybridized product was subjected to an interstrand copper-catalyzed Huisgen cycloaddition (CuAAC)

reaction to link the azide functionality of BB2 with the alkyne functionality of BB4, resulting in covalently linked HP5-HP3 strands, forming a semi-closed peptide macrocycle (**version 2** of DP-DEL) (**V**). This product was split again, subjecting the major part of the library **version 2** to Cys(oNv) group removal, allowing the scaffolds of both sub-libraries to connect through either a disulfide or through the addition of 3 different bis-electrophiles, reacting with both cysteine scaffolds simultaneously (**VI**). This resulted in fully-closed peptide macrocycles between the two **HP5/HP3** sub-libraries (**version 3** of DP-DEL). A final step of encoding followed, where DP-DEL **versions 1-3** could be encoded as building block 5 via hybridization of different codes 5 and a junction primer. Through T4 DNA polymerase and T4 ligase activity, the non-complementary encoding information of **HP5** and **HP3** sub-libraries + codes 5 could be transformed into a single continuous DNA strand. After this operation, all different versions of the DP-DEL were pooled to result in a finished master library.
**B** Characteristics of building blocks used for the creation of DP-DEL. Building blocks 1-3 consist of a selection of natural and non-natural amino acids and dipeptides. BB4 consists of alkyne carboxylic acids and BB5/code 5 define **versions 1-3** by varying the interaction between the two sub-libraries **HP5** and **HP3**. **C**) Distribution of molecular weights for the fully cyclized library members (**version 3** of library).

which were additionally derivatized by the use of different BB5 electrophile linkers (**version 3**) (Fig. 1)[75]. The different library versions were individually encoded with a fifth code and mixed together to create a final library mixture comprising 56,179,200 members.

## Affinity-based selections

**Control library selections.** DP-DEL performance was first assessed by testing the library against human carbonic anhydrase IX (CAIX) and SARS-CoV-2 non-structural protein 14 (NSP14), for which known

### A) Unselected Library

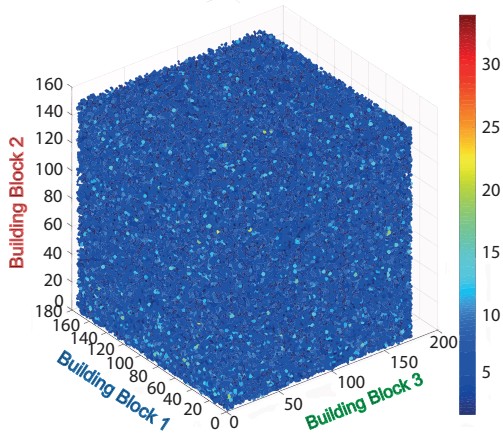

### B) CAIX Control Selections

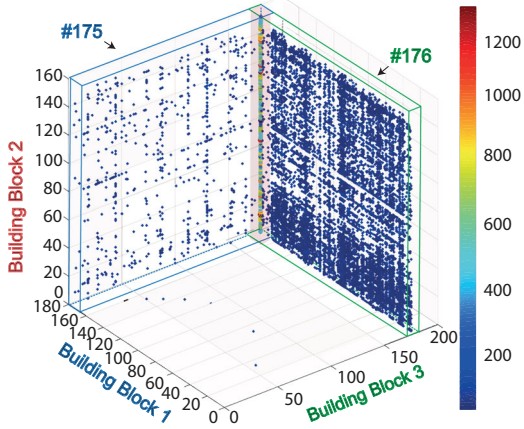

### C) NSP14 Control Selections

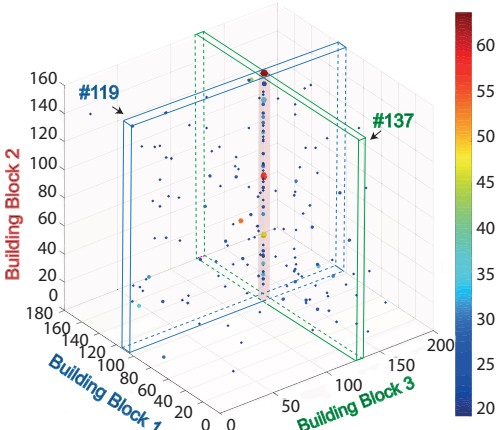

| Control Protein | Binding Substructure | Position |
|---|---|---|
| CAIX | | BB1: #175 |
| | | BB3: #176 |
| NSP14 | | BB1: #119 |
| | | BB3: #137 |

**Fig. 2 | High-throughput DNA sequencing fingerprints of naïve DP-DEL and control selections. A** High-throughput DNA sequencing (HTDS) results of unselected DP-DEL represented through a 3D plot displaying the distribution of the most prominent BBs 1-3 where each axis represents a BB position and each dot is a library member carrying a combination of said BBs, while the heat bar represents sequence counts. No cut-off was applied for unselected library. **B, C** Positive control concatenated triplicate HTDS against CAIX and NSP14 respectively. Spiked-in sulfonamide BB1-175 and BB3-176 showed preferential enrichment as binders for CAIX. Spiked-in S-adenosylhomocysteine BB1-119 and BB3-137 as a combination showed preferential enrichment as binders against NSP14. A cut-off of 20 counts was set for CAIX and NSP14. Selections were performed in triplicate and concatenated into a single dataset for visual display (for the selection results with the individual DP-DEL versions see Supplementary Information Fig. S28).

small-molecule binders exist[76–78]. DP-DEL included known binders against both proteins at two diversity element positions: 4-sulfamoylbenzoic acid (SABA) for CAIX and S-adenosylhomocysteine for NSP14, respectively, at both BB1 and BB3 positions.

The multi-dimensional selection fingerprints reflecting the occurrence of each library member after PCR amplification and high-throughput DNA sequencing[79], when reduced to 3-dimensional plots (BB1, BB2, BB3), showed a homogeneous distribution of library members in case of the unselected (naïve) DP-DEL (Fig. 2A), whereas selections against the two control targets clearly demonstrated a preferential enrichment of library members containing the known binders. This effect was strongly visible in the CAIX selections, where sulfonamide-possessing library members at BB1 and BB3 positions, respectively, formed enriched planes, while members possessing a sulfonamide at both BB1 and BB3 positions formed an even more pronounced vertical line (Fig. 2B). In the case of NSP14, only library members possessing S-adenosylhomocysteine at both BB1 and BB3 positions were enriched over the background (Fig. 2C).

**De novo selections with the DP-DEL and hit validation.** Following the validation of the DP-DEL on the control targets we aimed at testing the DP-DEL for de novo selections on further target proteins: We performed affinity-based selections against biotinylated thrombin, a blood coagulation factor and an important target of pharmaceutical interest for anticoagulant therapy[80]. Preferential enrichment was obtained for a series of library members originating from **version 2** of the library (Fig. 3A), hinting at the obtained peptide binders displaying a distinct preference for binders with a conformational restriction at the N-terminal extremity, while preferring less rigidity at the DNA-proximal C-termini. We decided to confirm this finding by hit resynthesis of the three selected binders of highest enrichment: 2;84;150;56;2, 2;84;148;56;2 and 2;35;78;112;2 (Fig. 3B). These hits were resynthesized on two complementary, fluorescently labelled locked nucleic acid (LNA) strands (**1-3**), such that they are presented in an analogous fashion to the library setting, and measured by fluorescence polarization. The LNA conjugates were tested as either N-terminally restrained (clicked) or unrestrained (open) compounds (Fig. 3B). **3** which was structurally markedly different from **1** and **2**, bound both in

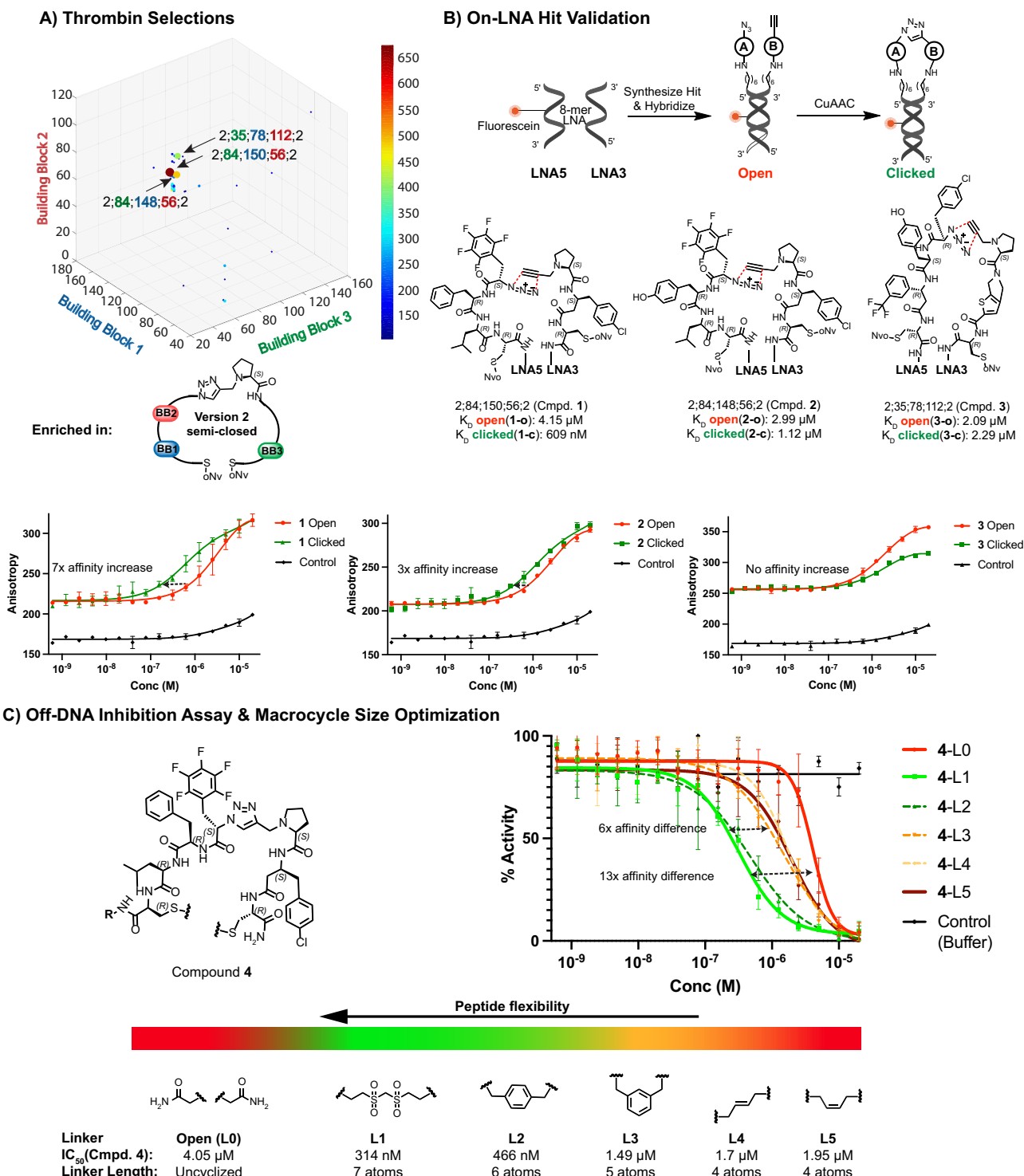

**Fig. 3 | DP-DEL thrombin selections. A** HTDS of concatenated triplicate selections against thrombin. Building blocks 1, 2 and 3 correspond to each respective axis. The heat bar represents the DNA sequence counts. The Plot is constrained to combinations from **version 2** of DP-DEL possessing BB4-position 2, due to the attainment of highest enrichment values from that version of the library (see Supplementary Information, section 5.4.2.). Arrows indicate the three most enriched library members from the selection. Cut-off: 100 counts. **B** On-LNA experimental validation of highest enriched library members **1-3** displayed on 8-mer locked nucleic acid (LNA) heteroduplexes with a fluorescein label by fluorescence polarization (FP). Conjugates were tested in open form approximating **version 1** of DP-DEL and clicked form approximates **version 2** of DP-DEL. Affinity values are given as a mean of triplicate experiments ($n = 3$), with the exception of **1-o**, which was measured as a duplicate ($n = 2$). **C** Off-DNA inhibitory measurements of selected compound **4** (R = Ahx-(H-Lys-NH₂)-5-carboxyfluorescein) against human alpha-thrombin, synthesized using a selection of linkers between the two cysteine scaffolds to test preferred peptide flexibility. L1-L5 allow the formation of variably sized macrocyclic peptides, while L0 yields a linear peptide. A heat bar shows the relation of $IC_{50}$ obtained and peptide steric constraint due to L0-L5 of binder. $IC_{50}$ values are obtained from experimental triplicates ($n = 3$) and given as a mean. Error bars indicate standard deviation of the replicates.

the clicked (**3-c**) and open (**3-o**) form (2.29 μM and 2.09 μM respectively), while **2** revealed a slight preference for the clicked (**2-c**) DP-DEL format (1.12 μM), compared with the open (**2-o**) format (2.99 μM). Strikingly, **1** displayed the highest affinity and a remarkable selectivity for the clicked (**1-c**) DP-DEL format (609 nM), compared with the open (**1-o**) format (4.15 μM) (Fig. 3B).

These encouraging results inspired the synthesis of **4** as the off-DNA derivative of **1** in its N-terminally clicked form, with the goal to test C-terminal peptide flexibility using different C-terminal linkers and compare it with the C-terminally open form. Compound **4** was cyclized with five bifunctional electrophilic linkers L1 - L5 to produce macrocycles with different flexibility, as well as kept in the C-terminally open configuration (L0), resulting in a linear peptide (Fig. 3C). The performance of each binder was assessed directly in an inhibition assay, demonstrating that the linear version of **4**-L0 was the weakest inhibitor (4.05 μM), whereas the most rigidified **4**-L5 performed marginally better (1.95 μM). Longer, more flexible linkers, however, gave rise to stronger inhibition, with the best compound, **4**-L1, featuring a bis(vinylsulfonyl)methane linker, with an IC$_{50}$ of 314 nM. Notably, these

findings were in full accordance with the obtained DP-DEL selection results, where **1** was strongly enriched only in the semi-closed DP-DEL version **2**, but neither in the rigid version DP-DEL version **3** nor in the linear DP-DEL version **1**.

Selections against streptavidin revealed two structurally novel, non-biotin related enriched library members 2;28;34;26;2 and 2;28;34;8;2, once more from the semi-closed DP-DEL version **2**, which were structurally similar (Fig. 4A). Resynthesis on LNA afforded the compounds **5** and **6** in open (**5-o**, **6-o**) and N-terminally clicked (**5-c**, **6-c**) format. Also, for streptavidin, conformationally restricted **5-c**, **6-c** resulted in enhanced affinities of 62 nM (for **5-c**) and 50 nM (for **6-c**), respectively, compared with 1.97 μM (for **5-o**) and > 2.5 μM (for **6-o**) in the open format, giving rise to a > 30-fold difference in affinity. **5-c** and **6-c** were then further investigated for C-terminal flexibility by synthesizing them off-DNA as fluorescein-conjugated derivatives **7** and **8**, respectively. Compound **7** was able to bind to streptavidin only with its C-terminally open, linear form with a K$_D$ of 173 nM, while compound **8** afforded a K$_D$ of 118 nM. Interestingly, all five C-terminally cyclized compounds **7**-L1 to **7**-L5 did not show any detectable binding (Fig. 4C).

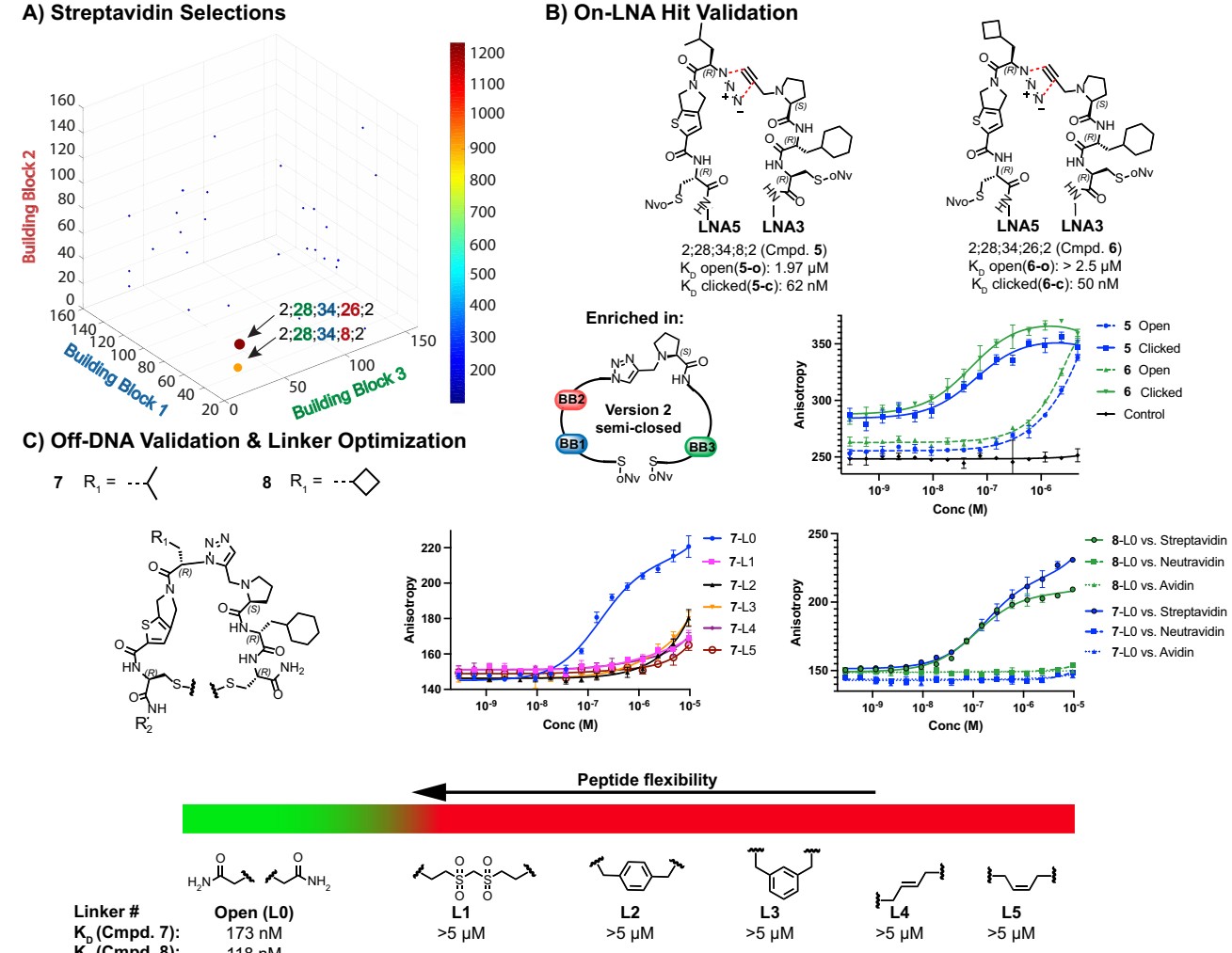

**Fig. 4 | DP-DEL streptavidin selections. A** DP-DEL HTDS of concatenated triplicate selections against streptavidin. Plot is constrained to combinations from version 2 of DP-DEL possessing BB4-position 2, due to the attainment of highest enrichment values from that version of the library (see supplementary section 5.4.2.). Arrows indicate the 2 highest enriched library members from the selection. Cut-off: 100 counts. **B** On-LNA experimental validation of highest enriched library members 5 & 6 displayed on 8-mer locked nucleic acid (LNA) heteroduplexes with a fluorescein label by fluorescence polarization (FP). Conjugates were tested in open (5-o, 6-o) form approximating version 1 of DP-DEL and clicked (5-c, 6-c) form approximates version 2 of DP-DEL. Affinity values are given as a mean of a duplicate or triplicate experiments. **C** Off-DNA affinity measurements of isolated streptavidin binders 7 & 8 (R2 = Ahx-(H-Lys-NH2)-5-carboxyfluorescein). Preferred peptide flexibility is examined through the synthesis of compound 7 using linkers L0-L5 and affinity of 8 is examined through synthesis with best performing L0. Selectivity of both 7-L0 & 8-L0 is tested against 3 different biotin-binding proteins, showing specificity towards streptavidin.

In addition, as the identified binders **7**-L0 and **8**-L0 were structurally different from biotin, a well-known high-affinity binder for streptavidin, avidin and neutravidin, we tested **7** and **8** for their potential to discriminate between these proteins. Remarkably, no binding of **7**-L0 and **8**-L0 was detectable for avidin and neutravidin, demonstrating the selectivity of our newly identified compounds for streptavidin.

As a third target we chose placental alkaline phosphatase (PLAP), an important biomarker for seminomas, endometrial, cervical, and ovarian cancer[81–83], which we and colleagues had already addressed by DEL selections with different libraries[84]. Our selections identified a series of β-homotyrosine analogs (BB1 position) coupled to phenylalanine analogs (BB2 position) from DP-DEL **version 1** (Fig. 5A). Hit resynthesis of the three highest enriched library members 2;16;81;47;1

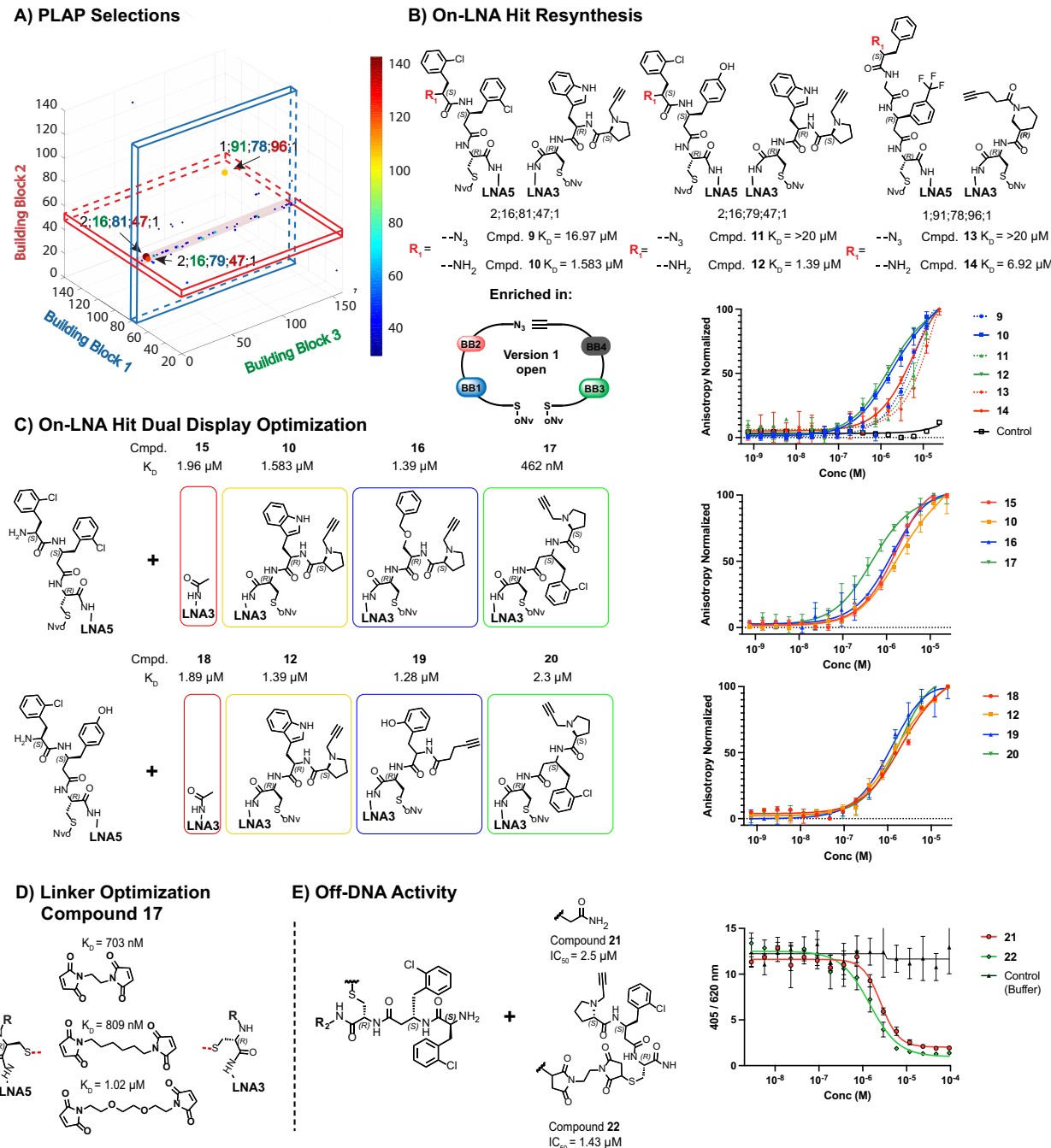

**Fig. 5 | DP-DEL PLAP selections. A** DP-DEL HTDS of concatenated triplicate selections against PLAP. Building blocks 1, 2 and 3 correspond to each respective axis. The heat bar represents DNA sequence counts. Plot is constrained to combinations from **version 1** of DP-DEL, for both BB4 positions (see supplementary section 5.4.2.). Arrows indicate the 3 highest enriched library members from the selection. Cut-off: 30 counts. **B** On-LNA experimental validation of highest enriched library members **9-14** displayed on 8-mer locked nucleic acid (LNA) heteroduplexes with a fluorescein label by fluorescence polarization (FP). Conjugates were compared in their **version 1** form with an azide group at the BB2 position or an NH$_2$ group. **C** Optimization of dual display partners for best performing binders **10** and **12**. Two additional enriched BB3 + BB4 combinations as well as a negative control were tested to improve on binders **10** and **12**. **D** Testing of different length maleimide linkers between cysteine scaffolds of both strands, to determine optimal spacing of best performing dual display partners **17**. **E** Off-DNA resynthesis and inhibition experiments of optimized binder **22** as well as its single display variant **21** (R$_2$ = Ahx-(H-Lys-NH$_2$)-FITC). Affinity and IC$_{50}$ values from experimental triplicates (n = 3) are given as a mean. Error bars indicate standard deviation of the replicates.

2;16;79;47;1 and 1;91;78;96;1 proceeded first on-LNA (Fig. 5B). During hit resynthesis, the diazotransfer reaction appeared to be unexpectedly challenging for the 2-chloro-L-phenylalanine building block, obtaining little conversion over the same reaction time used in library synthesis. This led us to believe that the free amine version of the conjugates was present during library selections as a truncate, and we decided to synthesize hence both the free amine, as well as the azide-containing conjugates (Fig. 5B). Fluorescence polarization assays confirmed the importance of the free amino group, where amine-containing conjugates **10**, **12** and **14** displayed $K_D$ values of 1.583 μM, 1.39 μM and 6.92 μM respectively, which were more than 10-fold better when compared to the respective azide-containing counterparts **9**, **11** and **13**.

Focusing on the best-performing conjugates **10** and **12**, the horizontal line in the selection fingerprints (Fig. 5A) indicates that binding enrichment primarily originates from the BB1 and BB2 positions in the **HP5** sub-library part. The minor contribution of the **HP3** sub-library to binding was confirmed with the acetate-capped conjugates **15** and **18**, where only little binding affinity was lost. We therefore prioritized optimizing the contribution of the **HP3** sub-library to the dual display system by testing several **HP3**-enriched building blocks, resulting in improved compound **17** with a $K_D$ of 462 nM. Off-DNA synthesis of the compound, using a bis-maleimide linker to connect the two complementary display parts, afforded compounds **21** and **22**. Interestingly, the PLAP-inhibition proceeded considerably better at lower pH (pH 9 compared with standard pH 10), indicating the importance of protonation of the free amino group (see Supplementary Information, section 7.2.2). Both compounds **21** and **22** inhibited PLAP with an $IC_{50}$ of 2.5 μM and 1.43 μM, respectively.

## Discussion

Peptides, and especially cyclic peptides, have recently experienced a resurgence as a therapeutic modality, since their large surface area may allow for binding also to challenging targets, and render them ideal disease-targeting ligands for the delivery of bioactive payloads such as radionuclides and cytotoxic drugs[1,85–87].

DEL technology offers a promising platform for the chemical assembly of peptide libraries. However, the macrocyclic libraries produced thus far by DEL have seen limited success, at least in part because of limitations imposed by poor library quality as well as an insufficient exploration of the conformational space. Additionally, the medicinal chemical optimization of selected macrocycles generally proves challenging, since structure-activity relationships of the individual building blocks as in the case of small-molecule DELs are harder to be derived due to the unpredictable eventual conformation of the macrocycle. To address these challenges, improving on library purity by utilizing the dual-display ESAC setup combined with the LED encoding, as well as exploring additional conformational space by producing the macrocycles in a two-step tunable process, rather than the usually employed one-step cyclization method, proved to be advantageous.

The LED method allowed us to synthesize a conformationally-tunable ESAC library displaying two diversity elements on each DNA strand ("2 + 2" setup). Unlike previous implementations of ESAC technology, which were focused on fragment display or affinity maturation, we used the approach to display two peptides in close proximity, which could subsequently be rigidified in a stepwise manner. This gradual introduction of conformational restriction by covalent linkage of the two peptides allowed us to display peptides with three different options for flexibility: open, semi-closed and closed macrocycle, while allowing us to encode each respective setup. Tuning of the macrocyclic size was further achieved by including both monomeric and dimeric building blocks as well as C-terminal linkers of varying size.

The combinatorial assembly allowed for an easy diversification from relatively small sub-libraries. The **HP5** sub-library, containing 26,600 members and the **HP3** sub-library, with 352 members, when mixed and combined into heteroduplexes, as well as later assembled according to 6 different conformations, achieved a final library diversity of 56 million unique library members.

In analogy to a solid phase-based DEL synthesis approach which we had recently developed[56] the modular ESAC system used in this work produces peptide DELs of high purity, since it distributes the split-and-pool synthesis on both strands (2×2 instead of 1×4 steps), allowing for individual compound purification after the first combinatorial synthesis step, while only the second synthesis cycle is performed "blindly" in a mixture. In addition, the dual-display system features the display of both constrained peptides and macrocycles in different configurations, facilitating the isolation of ligands with a preference for each of the three conformational configurations.

We discovered a nanomolar macrocyclic peptide inhibitor against thrombin, a target of pharmaceutical relevance, which might overcome the pharmacokinetic liabilities of larger peptide inhibitors[88–91]. In contrast, streptavidin showed a higher preference for more flexibility outlining the importance of peptide conformational adjustment. Additionally, our binder showed selectivity towards streptavidin versus the related biotin-binding proteins avidin and neutravidin, suggesting its use as a specific chemical probe[92].

Conversely, PLAP did not benefit from constraining the peptide flexibility, but rather preferred the flexible dual-display ESAC configuration, highlighting the utility of modularity and versatility in our setup. While PLAP has previously been explored by different DELs[84] the inhibitor identified here by dual-display is structurally novel and comprises simple and accessible building blocks.

In drug discovery, the identification of (typically low- to moderate-affinity) hits from large libraries constitutes a substantial bottleneck, and the option to easily explore different conformations of the same peptide within the very same library will greatly facilitate the identification of true hits. Looking ahead, we envision a further expansion of our design: crosslinking anchors could be inserted into the pairwise displayed peptides in between the N- or C-termini, which may enable the cyclization of peptides into macrocycles of various sizes and bicyclic structures. The strategy may also be well suited for exploring linkers of different lengths and geometries in conventional dual-display libraries[62,93]. Implementing further criteria for drug-likeness/cell permeability other than the inclusion of non-natural building blocks and cyclisation, such as restricting the number of H-bond donors and acceptors, as well as increasing lipophilicity through a careful selection of the utilized building blocks, may additionally prove valuable for subsequent lead development.

We believe that our approach will enable DEL technology to deeper investigate the underexplored cyclic peptide space and eventually provide ligands of high-affinity and high specificity also to targets so far considered as undruggable.

## Methods

Additional raw data, methods, procedures as well as characterization of compounds are described in the Supplementary Information.

### General on-DNA/LNA protocols

**DNA precipitation.** Dissolved DNA was precipitated via addition of 0.1x volume 5 M NaCl, followed by the addition of 3 volumes of ethanol. The mixture was then left in a −20 °C freezer overnight. The resulting suspension was centrifuged (25,000 x g, 4 °C) for 30 minutes. The supernatant was discarded, the pellet dried using speed vacuum and recovered as precipitated DNA.

**Scaffold attachment procedure.** Up to 1 μmol of DNA was dissolved in 800 μL MOPS buffer (50 mM, 0.5 M NaCl, pH 8). In a separate vial a

scaffold activation solution was prepared: 320 μL scaffold solution (100 mM, in DMSO) was mixed with 160 μL sulfo-NHS (100 mM, in 2:1 DMSO/water), 160 μL EDC (100 mM, in DMSO) and 720 μL DMSO. The activation solution was reacted for 30 min at 30 °C and then added to the DNA solution. The reaction was carried out overnight at RT and precipitated. The volumes were scaled accordingly when less DNA starting material was used.

**General amidation procedure (DMT-MM).** Up to 60 nmol of DNA was dissolved in 242 μL of borate buffer (85 mM, pH 9.3). To this solution 135 μL of acetonitrile was added, followed by 45 μL of carboxylic acid solution (200 mM, in DMSO). 28 μL of DMT-MM (300 mM, in water) was finally added and the reaction was shaken at RT for 2 hours. 28 μL of additional DMT-MM solution were then added and the reaction was shaken for further 2 hours and precipitated.

**Optimized dipeptide building block coupling procedure (EDC/ HOAt).** Up to 60 nmol of DNA was dissolved in 100 μL MOPS buffer (200 mM, 3 M NaCl, pH 8). In a separate vial 60 μL of 200 mM carboxylic acid in DMSO was mixed with 65 μL of EDC/HOAt/NMM (100:20:100 mM) in DMSO and 170 μL DMSO. The DMSO solution was immediately added to the DNA solution. The reaction was shaken for 1 hour and then a second batch of DMSO solution (60 μL carboxylic acid and 60 μL EDC/HOAt/ NMM) was added to the reaction, which was further shaken overnight and precipitated.

**Diazotransfer procedure.** Up to 100 nmol of DNA was dissolved in 20 μL water and added to a premixed solution of 20 μL $CuSO_4$ (10 mM, in water), 50 μL Imidazole-1-sulfonyl Azide.HCl (200 mM, in water) and 200 μL 50 mM $K_2CO_3$. The reaction was allowed to mix overnight at RT. The reaction was analyzed by LC-MS and precipitated.

**Interstrand click reaction.** To a pre-hybridized solution of up to 10 nmol of HP5 and HP3 strand were added 22 μL of TEAA buffer (1 M, pH 7), 32 μL DMSO, 25 μL $CuSO_4$/TBTA complex (1 mM, in 55% DMSO) and 25 μL sodium ascorbate (10 mM, in mQ). The reaction was carried out overnight at RT and then precipitated with NaCl.

**Scaffold deprotection and disulfide formation.** 1 nmol of HPLC-purified interstrand-clicked library was dissolved in 10 μL buffer consisting of sodium acetate (30 mM), ascorbic acid (10 mM) and semicarbazide hydrochloride (25 mM) pH 4.75. It was then placed under an UV lamp at 365 nm for 10-15 min at RT.

**Thioether formation.** Scaffold-deprotected DNA was dissolved in 35 μL $NH_4HCO_3$ buffer (60 mM, pH 8), followed by addition of 5.6 μL TCEP solution (10 mM, pH 7), and shaking for 1 hour at RT. 11.2 μL of bis-electrophile (10 mM) in acetonitrile was then added to 2.8 μL acetonitrile and 15.4 μL $NH_4HCO_3$ buffer. The reaction was mixed for 2 hours at 30 °C and precipitated.

**Fmoc deprotection procedure.** DNA was dissolved in 10% piperidine in water and the deprotection was carried out over 1 h. The reaction was then quenched using 1.5x volume of 3 M sodium acetate pH 4.5 and the DNA precipitated via addition of ethanol. Alternatively, the deprotection mixture was dried using a speed vacuum, redissolved in water and precipitated with NaCl.

**Encoding/Ligation procedure.** 1-50 nmol of DNA starting material was dissolved in water. 1.5 equivalents of coding strand and 2 equivalents of adaptor strand were added to make a total volume of 180 μL. The solution was heated to 80 °C and cooled to RT over 1 h. 20 μL of T4 DNA ligase buffer (10x, NEB) and 400U T4 DNA ligase (NEB) were then added and the ligation was incubated at 16 °C overnight.

**Terminal primer installation/polymerization of large encoding design.** Protocol was followed based on Plais et al. [69]. Briefly 1 nmol of premixed HP5 + HP3 step 2 library pools in any configuration (open, clicked, etc.) was mixed with 2.5 nmol junction primer and 2.5 nmol terminal primer in a total of 100 μL water. The strands were annealed by raising temperature to 95 °C for 10 min and cooled to room temperature. 50 μL of T4 DNA ligase buffer (10x, NEB), DNA polymerase (10U, NEB) and dNTPs (250 nmol) were added and mixture was kept at 16 °C for 30 min. T4 ligase was then added (400U, NEB) and ATP (300 nmol) and kept at 16 °C for a further 1 hour.

## General solid-phase/solution-phase synthesis procedures

**General solid-phase synthesis procedure.** Commercially available Tentagel® S NH2 base resin (Rapp polymere, # S30132) (500-1000 mg, 0.29 mmol/g loading) was swollen in DMF inside of a 10-25 mL fritted syringe. The resin was then manually modified with Fmoc-Rink Amide linker using general procedures for amide coupling and Fmoc deprotection. The resin was then used for hit resynthesis.

**Amide coupling.** DMF was added to free amino modified resin, followed by 2 equivalents of free carboxylic acid building block, 2 equivalents of HATU and 4 equivalents of DIPEA. The reaction was shaken for 2-12 hours and then quenched by removing the solution and washing the resin 5 times with DMF.

**Fmoc deprotection.** Resin was shaken for 1 hour with 20% Piperidine in DMF. The piperidine/DMF solution was then removed and the resin was washed 5 times with DMF.

**Alloc deprotection.** Resin was swollen in DCM. Alloc group at Lysine residue was treated with 0.2 equiv. $Pd(PPh_3)_4$ and 10 equiv. $PhSiH_3$ in dry DCM for 2-3 hours. The fritted syringe reaction vessel was vented every 10-15 min during the reaction. The resin was then washed 3x with DCM and 5x with DMF.

**Resin cleavage and purification.** The resin was shaken for one hour with a mixture of 95/2.5/2.5% trifluoroacetic acid/water/triisopropylsilane. The cleavage solution was then removed from the resin and stored. The procedure was repeated one additional time with fresh cleavage solution. The cleaved material was then either precipitated or diluted with MeOH and directly purified via RP-HPLC.

**Peptide precipitation.** 5-10 volumes of diethyl ether were added to the peptide cleavage solution. The resulting mixture was kept in a −20 °C freezer for 1-12 h and centrifuged. The crude peptide product could then be isolated as a pellet and dried.

**General procedure of CuAAC.** The resin was treated with 3 equiv. of CuI, 21 equiv. L-ascorbic acid and 1.2 equiv. of alkyne in 20% piperidine/ DMF for 20 h. The resin was then washed 5x with DMF.

**General procedure of azido-transfer.** The resin was swollen in DMSO. It was then treated with 3 equiv. of 1H-imidazole1-sulfonyl azide hydrochloride and 9 equiv. DIPEA in dry DMSO overnight. The resin was then washed 3x with DMSO and 5x with DMF.

**StBu deprotection and cyclization via thioether formation.** 1-2 mM hit intermediate possessing 2 Cys(StBu) amino acids in 50 μL DMSO was added to 350 μL DMF, 150 μL 1 M $NH_4CO_3$ pH8 and 50 μL of 160 mM TCEP in 1 M $NH_4CO_3$ pH8. The deprotection was carried out over 1 h and checked via LC-MS for completion. Upon completion 50 μL of 160 mM bis-electrophile in DMF was added. Reaction was then incubated further for 30 min and checked for completion via LC-MS. Reaction solution was then diluted with MeOH and directly purified via RP-HPLC

## Library synthesis

The **HP5** and **HP3** sub-libraries were created in parallel by coupling a Fmoc-L-Cys(oNv)-OH scaffold to respectively 17.5 μmol of a 5′-amino modified oligonucleotide (5′ NH2-C6-GGAGCTTCTGAATTCTGTGTGCTG [dSpacer][dSpacer][dSpacer][dSpacer]CTGGTCACTC 3′) for **HP5** and 3′-amino modified oligonucleotide (5′ Phos-AGTCACCTCA[dSpacer][dSpacer][dSpacer][dSpacer]CAGCACACAGAATTCA GAAGCTCC-C6-NH2 3′) for **HP3** sub-libraries using an EDC/sulfo-NHS method (Supplementary Information, section 4.2). Following Fmoc deprotection for both strands, a set of 175 Fmoc-protected amino acids and dipeptides was installed as BB1 on the HP5 sub-library strand (DMT-MM and EDC/HOAt methods, 60 nmol per BB) and a set of 176 Fmoc protected amino acids and dipeptides was installed as BB3 on the HP3 sub-library strand (DMT-MM and EDC/HOAt methods, 60 nmol per BB) (Supplementary Information, Tables S6 and S7). All conjugates were Fmoc deprotected, individually HPLC purified and encoded via enzymatic adaptor-mediated ligation. **HP5** sub-library strand conjugates were encoded utilizing codes 1 (5′ Phos-GTAGTCTCTC XXXXXX CTGTCGTACG 3′) and code 1 adaptor (5′ GAGAGACTACGAGTGACCAG 3′). **HP3** sub-library strand conjugates were encoded utilizing codes 3 (5′Phos-GAAGGGCTAC XXXXXX TTCGCTCGCT 3′) and code 3 adaptor (5′ TGAGGTGACTAGC GAGCGAA 3′). Ligation products were then pooled for each respective sub-library and the resulting pool was purified by HPLC at 60 °C (Supplementary Information, sections 4.3 and 4.4). The step 1 pool of the **HP5** sub-library was then re-split into 152 vessels (19 nmol each) and reacted with a second set of 152 Fmoc-protected amino acids and dipeptides as BB2 (DMT-MM and EDC/HOAt methods). The conjugates were then encoded utilizing codes 2 (5′Phos-TTGCTCACAC XXXXXXX GTCAACTC GGTCCTG 3′) and code 2 adaptor (5′ GTGTGAGCAACGTACGACAG 3′). The ligation products were subsequently pooled, Fmoc deprotected, subjected to a diazotransfer reaction and finally purified via HPLC at 60 °C to yield 350 nmol of a complete **HP5** sub-library (Supplementary Information, section 4.5). The step 1 pool of the **HP3** sub-library was split in 2 portions (250 nmol each) and encoded with codes 4 (5′-AGAAT CCTTGACGATCGATGG XXXXXXX TGAGTGAGTG-3′) and code 4 adaptor (5′ GTAGCCCTTCCACTCACTCA 3′). The ligation products then were reacted with a set of two alkyne carboxylic acids as BB4 (DMT-MM method) and treated with piperidine. The resulting conjugates were pooled and purified by HPLC at 60 °C to yield 284 nmol of a complete **HP3** sub-library (Supplementary Information, section 4.6). The two resulting sub-libraries were mixed stoichiometrically and hybridized to yield DP-DEL **version 1**. Part of the library product was then taken and subjected to an interstrand copper-catalyzed Huisgen cycloaddition (CuAAC), followed by a HPLC purification to yield DP-DEL **version 2** (Supplementary Information, section 4.7). Part of the **version 2** library product was finally and subjected to photo-deprotection at 365 nm of the Cys(oNv) scaffold. The resulting free thiols were allowed to form a disulfide or were reduced with TCEP and reacted with one of three different bis-electrophiles, simultaneously reacting with both cysteine scaffolds. The combination of these products yielded DP-DEL **version 3**. Each of the three library versions was individually encoded via different terminal primers (5′ ATCTGCATCAGTTCATGGGTA XXXXX CAGGACCGAGTTG AC 3′) and a junction primer (5′ Phos-GAGAGACTACGAGTGACCAGTTT GAGGTGACTAGCGAGCGAA 3′). Encoding was performed by hybridizing a terminal primer and junction primer to a respective sub-library, followed by treatment with T4 DNA polymerase, dNTPs and then T4 ligase, allowing the creation of a single continuous DNA strand, containing the full library encoding information (Supplementary Information, section 4.7). All DP-DEL versions could then be pooled to obtain a final library of 56,179,200 members.

## Affinity selections

DP-DEL was screened in triplicate against the target proteins of interest as previously reported[78]. Additional details are reported in Supplementary Information, section 5.

## Hit resynthesis

On-LNA hits were resynthesized on two complementary amino-modified LNA strands LNA5 (5′-(Fluorescein dT) AG TAG CC-3′) and LNA3 (5′-GG CTA CTA-3′). Hits were resynthesized on 50 nmol scale using on-DNA library synthesis DMT-MM and EDC/HOAt methods, as described in Supplementary Information, section 2.2. LNA conjugates were HPLC-purified and precipitated before being hybridized and tested. Some hybridized conjugates were subjected to an interstrand click reaction and purified via precipitation and an Amicon® Ultra Centrifugal Filter, cutoff 3 kDa, to remove any remaining salts/small molecules. A detailed description and characterization of all synthesized LNA conjugates is given in Supplementary Information, section 6.1.

Off-DNA hit resynthesis was performed on Tentagel® S NH2 base resin modified with a RinkAmide linker. Hits were resynthesized as fluorescein-linked conjugates using an Ahx-(H-Lys-NH₂) linker between the fluorescein and the resynthesized hit. All compounds were produced using standard solid-phase peptide synthesis procedures as described in Supplementary Information, section 2.3. A detailed account of compound resynthesis and characterization is given in Supplementary Information, section 6.2.

## Fluorescence polarization

LNA5 and LNA3 conjugates were hybridized by mixing 25 μL of LNA5 (2 μM) and 25 μL of LNA3 (2 μM) conjugates in water and heating to 70 °C for 5-10 min. After cooling down the solution to room temperature, 950 μL of 1x PBS was added to the LNA solution. The hybridized LNA conjugates were then tested via fluorescence polarization.

If the LNA conjugates were clicked together, they were diluted to 1 μM, 50 μL of a clicked conjugate was diluted with 950 μL 1x PBS and tested by fluorescence polarization.

Separately, a 16-point serial dilution of protein was performed in PBS in PCR tubes. 4 μL of the 50 nM conjugate was mixed with 4 μL of each protein dilution in black 384-well plates to a final volume of 8 μL. The plate was incubated for 30 min in the dark and centrifuged briefly to remove bubbles. Fluorescence anisotropy was measured at 535 nm. Data was fitted using Prism 10. Raw data from experimental FP measurements can be found in Supplementary Information, section 7.1.

## Inhibition assays

Thrombin inhibition protocol was adapted from Habeshian et al.[4]. A 16-point dilution series of inhibitor was performed starting at 33 μM in Tris buffer (100 mM Tris-Cl, 150 mM NaCl, 10 mM MgCl₂, 1 mM CaCl₂, 2% DMSO). 9 μL from each dilution point was mixed with 3 μL thrombin solution (9 nM, in Tris buffer) and 3 μL Z-Gly-Gly-Arg-AMC solution (225 μM, in Tris buffer) in black 384-well plates. Fluorescence (excitation at 360 nm, emission at 465 nm) was measured every 3 min for 30 min. The slopes were normalized & fitted, in order to calculate IC50 using Prism 10. Data was measured in triplicate. Raw data from thrombin measurements can be found in Supplementary Information, section 7.2.1.

For the PLAP inhibition assay, 500 μM of inhibitor was used for a 16-point serial dilution in 10% DMSO/activation buffer in PCR tubes. Separately, a 12.5 nM solution of PLAP and a 4.5 mM solution of p-nitrophenylphosphate (pNPP) were prepared in activation buffer (see below). 8 μL of each inhibitor dilution was mixed with 16 μL of enzyme solution and 16 μL of pNPP solution and gently mixed. The assay was allowed to develop in the dark for 30 min (activation buffer pH10: 150 mM diethanolamine, 1 mM MgCl2, 60 mM ZnCl2 in water, pH 9.8) or 1 hour (activation buffer pH9: 150 mM Tris, 1 mM MgCl2, 60 mM ZnCl2 in water, pH 9.0). 50 μL of 2 M NaOH was then added to quench the reaction. Readout was performed to detect 405 nm vs 620 nm absorption. Data was fitted using Prism 10. Raw data from PLAP measurements can be found in Supplementary Information, section 7.2.2.

**Reporting summary**

Further information on research design is available in the Nature Portfolio Reporting Summary linked to this article.

## Data availability

The main data related to the study are given in the Article, its Supplementary information and from corresponding author(s) upon request. Raw data relating to experimental affinity/inhibition measurements is listed in section 7 of the supplementary information document. Information relating to encoding and building blocks included in the library is listed in section 4 of the Supplementary information. Source data are provided with this paper.

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

## Acknowledgements

We are grateful to N. Ban, and D. Yudin and M. Jia from the lab of N. Ban for the help in cloning and expression of NSP14 protein. We thank T. Georgiev for providing protocols and valuable guidance to setup PLAP inhibition assays. We thank S. Oehler and N. Favalli for helpful scientific discussions and advice. We thank F. Migliorini and the Philochem chemistry team for help in purification of the first step of library synthesis. We thank G. Assoni and S. Zhong for helpful suggestions. We thank G. Hoppeler for calculation of physicochemical properties of the library. We acknowledge the Functional Genomics Center Zurich for conducting high-throughput DNA sequencing. Instant JChem (ChemAxon) was used for the structure and data management (http://www.chemaxon.com). J.S. gratefully acknowledges financial support from ETH Zürich and Innosuisse – Swiss Innovation Agency (grant 48350.1 IP-LS to J.S.).

## Author contributions

D.P. and J.S. designed the project. D.P. constructed the DEL libraries. L.P. and D.P. worked on the LED encoding. K.S., J.C., M.K. and A.G. cloned and expressed proteins, D.P. and A.G. performed DEL selections and analysed the results. A.L. performed data analysis. G.B., S.C. and D.N. provided advice and helped with hit validation and data analysis. D.P., A.G. and J.S. wrote the manuscript with contributions and corrections from all authors.

## Funding

## Competing interests

D.P., M.K., A.G. and J.S. are assignees of a patent application WO2022/084486. D.N. is co-founder and shareholder of Philochem AG (http://www.philochem.com), a company active in the field of DELs. The other authors declare no competing interests.
