## [Transparent Peer Review file · Nature Communications]

Flexibility-tuning of dual-display DNA-encoded Chemical Libraries facilitates cyclic peptide ligand discovery

Corresponding Author: Professor Jörg Scheuermann

Version 0:

Reviewer comments:

Reviewer #1

(Remarks to the Author)

A great problem in the encoded chemistry field is the deteriorating quality of large DELs as syntheses get longer. This problem is especially problematic for peptide DELs because the synthetic routes are often several steps. The work outlined in this paper presents a fresh approach where two separate libraries are created (one 3' and one 5') and these are then combined to create a cross-over library of the two sub-sets. These libraries are then further modified by cyclization at the termini and then ring rigidification by an additional stapling reaction. An important feature of the present work is that all three forms of the assembled library are tested separately, allowing a systematic study of the impact of cyclization. There is a lot of work here and it is all very well executed and convincingly characterized. After addressing my critiques below I think this paper will be ready for publication:

1. there is extensive discussion of the importance of purity, but the final cyclized compounds often show several overlapping peaks in the HPLC (for example, the two cases in the SI p. 27). Could the authors comment on this.
2. There are several other recently developed peptide macrocycle approaches, either through nanoscale synthesis (Heinis), or through PNA display (Winssinger) (the ribosome reprogramming of Suga is well discussed in the intro). I think it would be important for the authors to give a more careful comparison to these other approaches. Again especially the Heinis nanoscale synthesis work and the Winssinger suprabody PNA work are especially important to contextualize here.
3. The way the general structure is shown in Fig. 2C is confusing and should be reconsidered. A single general structure with a neighboring legend with details and numbering is one possible alternative.
4. The fact that some targets/hits preferred flexibility, while others preferred rigidification is interesting, and highlights the importance of having careful models where each possibility is tested. In this vein I would like to see an unfiltered X/Y plot of open vs. closed selections so we can see how often the same hits are recurring, as well as how often cyclization/stapling leads to new hit series emerging.

Reviewer #2

(Remarks to the Author)

The manuscript by Scheuermann and colleagues describes a novel strategy to use DNA-encoded libraries for the screening of macrocyclic libraries. The advantage of DEL compared to other "biologically" encoded screening platforms is the possibility to incorporate a large variety (and high content) of non-proteinogenic amino acids. Searching for novel scaffolds that combine high target affinity with cell permeability, this may be of central importance. The hallmark of this elegant study is the combination of two independent libraries that are combined in three different ways to provide scaffolds with different topologies. The results of the study are well presented and a large and convincing amount of data is presented. This reviewer considers the manuscript suitable for publication in Nat. Commun. after the following points have been addressed.

- 1) How was the library designed? Did drug likeliness serve as a criterion (e.g. small number of H-bond donors and number of rotatable bonds)? This should be introduced in more detail.

2) The authors highlight the importance of tuning flexibility for the generation of high-affinity binders. The approach modulates flexibility at two different levels: i) topology (open vs. semi-closed vs. closed) and ii) variation of C-terminal linkers to replace the templating DNA strands. While i) is an elegant feature, ii) is a necessity since DNA has to be replaced to provide useful inhibitors. This aspect should be clearly stated and when discussing the “tuning” of flexibility it should be distinguished between these two levels.

3) The authors order compounds by “peptide flexibility”. While this order may apply for the linker architecture used to replace the DNA tags, it is unlikely that the flexibility of the entire macrocycle is indeed fully determined by crosslink flexibility. It is possible that polar or aromatic groups in the crosslink interact with groups within the macrocycle and therefore increase rigidity. The authors should revisit the wording here or look into this in more detail. This is an important aspect as the manuscript focuses on “flexibility-tuning”.

4) Minor: Figure 3C and 4C bottom: Format of the two “NH₂” groups should be corrected.

Reviewer #3

(Remarks to the Author)

Peptide and peptide-like macrocycles have undergone a resurgence of interest in the pharma industry, spurring new efforts in the making the discovery of macrocyclic protein ligands more reliable. By far, the most powerful methodology is mRNA display, but this technique relies on ribosomal synthesis of the macrocycles, which limits the type of building blocks that can be used. DEL offers an alternative, since one can use whatever type of unnatural amino acid one wishes. However, a major limitation of current methodology is that DELs constructed using more than 2-3 steps (called “cycles”) are generally of low quality with many impurities. Thus, to make DELs of peptide macrocycles with multiple sites of variability is currently impractical.

This paper by Scheuermann and colleagues reports a strategy to address this issue, at least in part. They apply their previously reported ESAC (Encoded self-assembling chemical) library technology to the creation of DNA-encoded libraries (DELs) of peptide macrocycles. ESAC, developed originally for the discovery of protein-binding fragment pairs, involves creating two separate DELs that contain a short complementary DNA sequence. This allows the two library components to come together when mixed. In this case, each piece of the library contains two variable amino acids. Thus, each library is a “two cycle” DEL, which can be made very efficiently. A new twist on the ESAC library strategy is that the separately synthesized pieces contain moieties that allow cyclization on the N- or C-terminus of the paired linear molecules. By linking one, two, or none, of these linkages, the authors can use the same constructs to create essentially three sub-libraries that differ in the degree of stiffness of the macrocycle.

They use the libraries created in this fashion to identify ligands for three different proteins. Not surprisingly, the stiffness of the macrocycle can have significant effects on the binding affinity.

All in all, this is an interesting, technically sound, study. It will be of interest to all practitioners of DEL chemistry. It deserves to be published in a good journal such as Nature Communications. A limitation of this technology is that two of three sub-library formats are dependent on the DNA linkage holding the two pieces of the ligand together. This must be substituted with a synthetic linker, which introduces an additional element of optimization that must be done post-screening. It would have been interesting for the authors to “go the extra mile” and develop a combinatorial second screening step to identify high affinity macrocycles in which a DNA-supported linkage is no longer necessary, for example by making the peptidic part of the molecule prior to DNA encoding, then encoding a large library of linkers. Nonetheless, given the paucity of methods to make DELs of peptide macrocycles, this study is an excellent addition to the literature.

Version 1:

Reviewer comments:

Reviewer #1

(Remarks to the Author)

the authors have discussed and carefully considered every point, making changes where needed. In my view the manuscript is now ready for publication.

Reviewer #2

(Remarks to the Author)

All reviewer comments have been adequately addressed by the authors.
Publish as is.

Re: Point-by-Point Response to the Referees' Comments

Dear Reviewers,

We would like to thank you for your kind review and insightful comments. Below, we indicate how we have addressed the points raised by you. Our answers and the changes in the manuscript are given in blue.

Reviewer #1:

A great problem in the encoded chemistry field is the deteriorating quality of large DELs as syntheses get longer. This problem is especially problematic for peptide DELs because the synthetic routes are often several steps. The work outlined in this paper presents a fresh approach where two separate libraries are created (one 3' and one 5') and these are then combined to create a cross-over library of the two sub-sets. These libraries are then further modified by cyclization at the termini and then ring rigidification by an additional stapling reaction. An important feature of the present work is that all three forms of the assembled library are tested separately, allowing a systematic study of the impact of cyclization. There is a lot of work here and it is all very well executed and convincingly characterized. After addressing my critiques below I think this paper will be ready for publication:

We are grateful to the referee for his/her nice comment on our work.

1. there is extensive discussion of the importance of purity, but the final cyclized compounds often show several overlapping peaks in the HPLC (for example, the two cases in the SI p. 27). Could the authors comment on this.

We thank the reviewer for his comment on some of the LC-MS presented in the Supplementary information. The respective compounds were obtained in multiple steps: first the respective (pure) two compounds of the sub-libraries were hybridized and clicked, this was monitored by LC-MS (see SI p. 26) and showed a pure conjugate of both sub-libraries. Identification of the ca. doubled the mass by deconvolution is already very delicate in this mass range (see the clean LC-MS of the thiol-protected compound). Afterwards, the lower ring-closure was performed by photocleaving the Cys protecting groups and reacting with the bifunctional electrophiles. Although, as the reviewer mentioned the corresponding LC trace of the MS does not look perfect, we could show in the deconvoluted mass spectra (corresponding to the sum of the whole imperfect peaks - which in the LC might still contain UV-absorbing residual electrophile or cleaved protecting group) of these compounds both that the precursor was no longer detectable and that basically only the desired conjugate was deconvoluted.

2. There are several other recently developed peptide macrocycle approaches, either through nanoscale synthesis (Heinis), or through PNA display (Winssinger) (the ribosome reprogramming of Suga is well discussed in the intro). I think it would be important for the authors to give a more careful comparison to these other approaches. Again especially the Heinis nanoscale synthesis work and the Winssinger suprabody PNA work are especially important to contextualize here.

We thank the reviewer for his valuable comment. We fully agree with her/his suggestion and have now adjusted our Introduction in order to contextualize the work of C. Heinis and N. Winssinger in detail, providing also the respective references.

3. The way the general structure is shown in Fig. 2C is confusing and should be reconsidered. A single general structure with a neighboring legend with details and numbering is one possible alternative.

We agree with the reviewer and have now adjusted Fig.2C for more clarity.

4. The fact that some targets/hits preferred flexibility, while others preferred rigidification is interesting, and highlights the importance of having careful models where each possibility is tested. In this vein I would like to see an unfiltered X/Y plot of open vs. closed selections so we can see how often the same hits are recurring, as well as how often cyclization/stapling leads to new hit series emerging.

We thank the reviewer for his valid question and we agree that providing this information would further clarify our findings. Therefore, we now provide unfiltered X/Y plots of the selections against all targets CAIX, NSP14, streptavidin, thrombin and PLAP as SI figures S27 and S28. Additionally, a new Table S14 is now added which shows the differential enrichment of the main hits according to library cyclization version. To improve readability, we have also better annotated the hits in the selection fingerprints of the Supplementary Figs. S27, S28 and S29.

Reviewer #2:

The manuscript by Scheuermann and colleagues describes a novel strategy to use DNA-encoded libraries for the screening of macrocyclic libraries. The advantage of DEL compared to other “biologically” encoded screening platforms is the possibility to incorporate a large variety (and high content) of non-proteinogenic amino acids. Searching for novel scaffolds that combine high target affinity with cell permeability, this may be of central importance. The hallmark of this elegant study is the combination of two independent libraries that are combined in three different ways to provide scaffolds with different topologies. The results of the study are well presented and a large and convincing amount of data is presented. This reviewer considers the manuscript suitable for publication in Nat. Commun. after the following points have been addressed.

We thank the reviewer for this kind assessment.

1. How was the library designed? Did drug likeliness serve as a criterion (e.g. small number of H-bond donors and number of rotatable bonds)? This should be introduced in more detail.

The reviewer raises a valid point. In this study, our approach to implement drug-likeness of the presented compounds was a) the incorporation of a high number of unnatural and D-amino acids, b) incorporating the non-natural triazole ring and c) the possibility of cyclization. The main aim of our work was to obtain larger macrocycle sizes (using the ESAC design and also dipeptide building blocks) at a high purity, which is currently not achievable with standard DEL technology. In further iterations of DEL syntheses with the concept proposed here, further criteria for drug-likeness, such as a defined number of H-bond donors/acceptors and rotatable bonds, may certainly be implemented. We have modified the Discussion to stress this important point.

2. The authors highlight the importance of tuning flexibility for the generation of high-affinity binders. The approach modulates flexibility at two different levels: i) topology (open vs. semi-closed vs. closed) and ii) variation of C-terminal linkers to replace the templating DNA strands. While i) is an elegant feature, ii) is a necessity since DNA has to be replaced to provide useful inhibitors. This aspect should be clearly stated and when discussing the “tuning” of flexibility it should be distinguished between these two levels.

The reviewer is fully right, it is a necessary step to cyclize the compounds “at the bottom” via bifunctional electrophilic crosslinkers to mimic the non-covalent hybridization provided by the dual-display on DNA. In the case of the closed DEL version the linker-mediated closure is provided before selection on the target, and the flexibility here is provided by the varying bis-electrophiles. In the case of the open and semi-closed DEL, this “fine-tuning” is indeed provided post selection by replacing the DNA with the linkers. We have therefore adjusted the text to further clarify these levels of fine-tuning.

3. The authors order compounds by “peptide flexibility”. While this order may apply for the linker architecture used to replace the DNA tags, it is unlikely that the flexibility of the entire macrocycle is indeed fully determined by crosslink flexibility. It is possible that polar or aromatic groups in the crosslink interact with groups within the macrocycle and therefore increase rigidity. The authors should revisit the wording here or look into this in more detail. This is an important aspect as the manuscript focuses on “flexibility-tuning”.

We agree with the reviewer that intramolecular interactions between the side chains (e.g., pi-pi stacking or H-bond bridging) may additionally influence macrocycle rigidity. These effects, however, are difficult to predict at the DEL design stage and will, in a very large DEL, arise serendipitously. We have mentioned this now explicitly in the Introduction.

4. Minor: Figure 3C and 4C bottom: Format of the two “NH₂” groups should be corrected.

Many thanks for spotting this. We have now corrected the Figures.

Reviewer #3:

Peptide and peptide-like macrocycles have undergone a resurgence of interest in the pharma industry, spurring new efforts in the making the discovery of macrocyclic protein ligands more reliable. By far, the most powerful methodology is mRNA display, but this technique relies on ribosomal synthesis of the macrocycles, which limits the type of building blocks that can be used. DEL offers an alternative, since one can use whatever type of unnatural amino acid one wishes. However, a major limitation of current methodology is that DELs constructed using more than 2-3 steps (called “cycles”) are generally of low quality with many impurities. Thus, to make DELs of peptide macrocycles with multiple sites of variability is currently impractical.

This paper by Scheuermann and colleagues reports a strategy to address this issue, at least in part. They apply their previously reported ESAC (Encoded self-assembling chemical) library technology to the creation of DNA-encoded libraries (DELs) of peptide macrocycles. ESAC, developed originally for the discovery of protein-binding fragment pairs, involves creating two separate DELs that contain a short complementary DNA sequence. This allows the two library components to come together when mixed. In this case, each piece of the library contains two variable amino acids. Thus, each library is a “two cycle” DEL, which can be made very efficiently. A new twist on the ESAC library strategy is that the separately synthesized pieces contain moieties that allow cyclization on the N- or C-terminus of the paired linear molecules. By linking one, two, or none, of these linkages, the authors can use the same constructs to create essentially three sub-libraries that differ in the degree of stiffness of the macrocycle.

They use the libraries created in this fashion to identify ligands for three different proteins. Not surprisingly, the stiffness of the macrocycle can have significant effects on the binding affinity.

All in all, this is an interesting, technically sound, study. It will be of interest to all practitioners of DEL chemistry. It deserves to be published in a good journal such as Nature Communications. A limitation of this technology is that two of three sub-library formats are dependent on the DNA linkage holding the two pieces of the ligand together. This must be substituted with a synthetic linker, which introduces an additional element of optimization that must be done post-screening. It would have been interesting for the authors to “go the extra mile” and develop a combinatorial second screening step to identify high affinity macrocycles in which a DNA-supported linkage is no longer necessary, for example by making the peptidic part of the molecule prior to DNA encoding, then encoding a large library of linkers. Nonetheless, given the paucity of methods to make DELs of peptide macrocycles, this study is an excellent addition to the literature.

We thank the reviewer very much for her/his precise analysis of our work and for the great suggestion to construct also an encoded library of linkers. While we consider this (secondary) library out of the scope of the present manuscript, we would like to state that we have started working in this direction. However, linker optimization by DEL in this case would constitute a secondary selection and naturally be limited to only a few of the hit molecules obtained from the primary DEL. The focus of this work was the comparison of the different cyclization formats, yet we are certain that future work from our labs and others will address further aspects of macrocycle flexibility in greater detail.

Once again, we would like to thank you for reviewing our manuscript.

We feel that we could address all the points raised and we hope that our revised manuscript may now be acceptable for publication.